# Do People Judge Sexual Harassment Differently Based on the Type of Job a Victim Has?

**DOI:** 10.3390/bs15060757

**Published:** 2025-06-01

**Authors:** Carolyne Georgiana Halfon, Destiny McCray, Danica Kulibert

**Affiliations:** Department of Psychological Science, Kennesaw State University, Kennesaw, GA 30144, USA; chalfon25@gmail.com (C.G.H.); dmccra13@students.kennesaw.edu (D.M.)

**Keywords:** sexual harassment, prototypes, gender, gender norms

## Abstract

Victims of sexual harassment report facing barriers and retaliation for reporting their sexual harassment. The current study assessed one potential reason for these issues: perceptions of sexual harassment events. Participants (N = 427) read about different sexual harassment events and were told that the woman in the event worked with a masculine (e.g., EMT, school police officer, plumber) or feminine job (e.g., nurse, teacher, cleaning staff). Across three different sexual harassment claims (e.g., unwanted romantic attention, physical groping, being shown sexually explicit images), participants reported that women in masculine jobs were less prototypical women than women in feminine jobs. Furthermore, these perceptions of prototypicality impacted how participants viewed the victim’s sexual harassment event. The less a person views a victim as a typical woman, the less likely they are to label the victim’s experience as sexual harassment. The results suggest that perceptions of sexual harassment are directly impacted by how people view a victim. Implications around prototypes of women and sexual harassment claims are discussed.

## 1. Do People Judge Sexual Harassment Differently Based on the Type of Job a Victim Has?

Sexual harassment is broadly defined as any unwelcomed behavior of a sexual nature that affects the dignity of individuals at work ([20]). Sexual harassment encompasses a range of unwelcome behaviors, from verbal remarks to physical advances, that create a hostile work environment ([22]). Sexual harassment is also a common occurrence, with 25% to 85% of women reporting having experienced some form of sexual harassment in the workplace ([20]; [45]). This wide range reflects the challenges in measuring and reporting sexual harassment, highlighting both the variability in personal thresholds and the prevalence of the issue across different industries and job types ([20]). For example, [37] ([37]) reported that 56% of women nurses had experienced sexual harassment in the workplace, but [56] ([56]) reported 80% of women medical residents experienced gender-based sexual harassment, and 43% experienced unwanted sexual or romantic attention. Specific career paths can also create barriers around reporting and addressing sexual harassment; women police officers report threats of retaliation and dismissive attitudes towards sexual harassment in the workplace ([16]). Furthermore, [50] ([50]) theorize that women working in male-dominated fields may be more likely to experience sexual harassment due to the increased sexualization and feminization of women in those fields. Addressing the widespread issue of sexual harassment requires a comprehensive understanding of the factors that influence the perception of sexual harassment incidents, one of which may be the workplace setting or occupation.

Occupational differences may impact perceptions of sexual harassment, with gender norms associated with various job types potentially leading to the minimized recognition of harassment claims. Individuals in job roles that challenge traditional gender stereotypes—such as men in caregiving positions or women in leadership roles—may find their claims perceived as less legitimate or entirely unrecognized compared to individuals in gender-typical roles ([34]; [7]; [21]). Skepticism toward victims who deviate from societal expectations can be attributed to prototype theory ([51]), which suggests that people have mental representations of typical victims, leading to doubt when individuals fall outside these prototypes ([29]).

### 1.1. Gender Norms and Perceptions of Sexual Harassment

Gender norms refer to the societal normalization of expectations and rules that dictate appropriate behaviors, roles, and attributes for individuals based on their perceived gender ([26]). These norms are deeply embedded in culture and influence how people perceive, interpret, and react to others’ actions and identities ([15]; [4]; [26]; [17]). Gender norms significantly shape prototypes in the context of workplace sexual harassment. Traditional gender norms suggest that women are more likely to be viewed as weak or passive, and men are more likely to be viewed as dominant or assertive ([27]; [41]). As a result, women who conform to gender norms—such as young, white, feminine women—are more readily perceived as prototypical women and prototypical victims of harassment ([39]; [41]). On the other hand, those who deviate from these gender norms—such as unattractive, older, black, or masculine women— are less likely to fit the prototypical woman or prototypical victim of sexual harassment, and their experiences may be minimized ([49]; [8]; [29]; [10]; [58]). Perceptions of women in the workplace may also relate to the field the woman is working in.

Societal expectations about appropriate behavior for men and women in the workplace influence how we interpret their actions and experiences ([19]; [41]). For instance, a woman in a male-dominated engineering environment might face overt harassment that is normalized or dismissed as part of the culture ([61]). Attempts to report or address the behavior could be met with resistance or disbelief, with colleagues rationalizing the behavior as “part of the job” or suggesting she lacks resilience or team spirit ([6]). Research shows that sexual harassment in male-dominated workplaces is often more normalized, with lower recognition of such incidents for women, whereas men in female-dominated roles may experience disbelief or minimization of their experiences, as these roles challenge societal expectations about masculinity and strength ([34]; [7]).

Gendered stereotypes not only influence the likelihood of recognizing sexual harassment but also shape judgments about the severity of the incident and the credibility of the victim ([39]). [3] ([3]) findings revealed that men and women differed in their recognition of what constituted sexual harassment, with women being more likely to label certain behaviors as sexual harassment compared to men. Furthermore, the study indicated women in traditionally feminine roles were more likely to be viewed as credible victims of harassment, whereas those in masculine roles faced skepticism regarding their experiences; the study supports the claim that gender norms may impact the recognition of harassment and shape judgments about its validity and the victims.

### 1.2. Prototypes and Perceptions of Sexual Harassment

Prototype theory, borrowed from cognitive psychology, suggests that people categorize experiences based on typical examples or “prototypes” that embody the most common attributes of a category ([51]). For example, a prototype of an apple would be red, hard, sweet, and spherical, although apples can also be yellow, orange, pink or green and sour or tart and oval, lopsided, or cut, etc. In the context of sexual harassment, prototypes of victims often include characteristics such as the victim’s gender, age, appearance, and behavior ([28]; [29]; [39]). Traditionally, people tend to also believe romantic or sexual attraction is related to sexual harassment incidents ([5]; [30]). Previous research has shown that when victims deviate from the expected prototype—attractive, young, feminine, and white—their experiences are less likely to be perceived and recognized as sexual harassment ([2]; [29]; [39]). In their study, [29] ([29]) investigated how these limited prototypes can lead to disregarded and neglected victims, showing that individuals who deviate from these conventional attributes may be dismissed when they report harassment due to their deviation from an observer’s prototype. For example, older women or those occupying non-traditional gender roles are often not seen as typical victims, leading to their claims being dismissed or overlooked ([29]; [9]). These perceptions and prototypes of victims could result in inadequate support for victims and insufficient corrective actions within the workplace.

Prototypes, by establishing preconceived ideas about who is most likely to be a victim or perpetrator, lead to a disparity in the recognition and response to incidents based on the victim’s conformity to gender norms ([39]). For example, a mismatch between societal expectations and the victim’s characteristics can hinder bystander intervention and reduce the likelihood that their experiences are acknowledged or supported, which can increase the barriers racial minority victims of sexual harassment experience ([55]; [47]; [58]). Prototype theory highlights how people categorize victims of sexual harassment based on typical attributes, such as gender, age, race, and appearance. For example, racial biases can have an impact on claim validity; in other research, people were more prone to perceive White, rather than Black, women as sexual harassment victims ([2]; [39]). Prototype theory suggests that the tendency to dismiss Black victims of sexual harassment more than White victims may stem from a mismatch between the societal prototype of a sexual harassment victim being a “typical woman” (i.e., a White woman; [29] and the race of the specific victim (i.e., a Black woman). However, these prototypes do not exist in isolation; they are shaped by broader societal constructs, with gender norms playing a predominant role in the context of sexual harassment.

Although substantial research has examined the frequency and psychological impacts of sexual harassment ([13]; [22]), there is a gap in the literature regarding how occupational roles influence the perception and validation of harassment claims. The influence of job type on perceptions of sexual harassment is critical because job roles are often infused with gender-based stereotypes and are gender-typed as masculine or feminine based on societal norms and expectations ([34]). Masculine jobs are typically associated with traits such as assertiveness and physical strength, whereas feminine jobs are linked to nurturing and supportive characteristics ([36]; [24]; [19]; [18]). These gendered expectations extend beyond job functions and shape societal attitudes toward individuals in those roles, influencing perceptions of workplace behavior and experiences. For instance, women in traditionally masculine roles, such as STEM fields or law enforcement, may be perceived as deviating from gender norms ([52]), which could affect whether their claims of sexual harassment are taken seriously ([7]).

Job roles also evoke societal assumptions about gendered behaviors and affect judgments about whether a woman’s experiences are consistent with cultural understandings of victimhood ([35]). Research by [25] ([25]) supports the idea that women in more feminine roles are often viewed through the lens of benevolent sexism, where they are seen as more fragile or deserving of protection. In contrast, women in masculine roles might be subjected to hostile sexism, where they are judged harshly for stepping outside traditional gender norms and gender roles. Understanding how occupational context influences perceptions of sexual harassment is essential for developing effective workplace policies. If sexual harassment claims from women in masculine roles are less likely to be believed or validated, current policies and training on sexual harassment may fail to appropriately address sexual harassment for these victims. By promoting awareness of gender norm-based biases, research can help ensure all sexual harassment claims are taken seriously, regardless of the victim’s job or gender ([34]).

### 1.3. Current Study

Recognizing that sexual harassment occurs across diverse job settings, this study’s objective is to examine whether the nature of a job—classified as either traditionally feminine (e.g., housekeeping, teaching, nursing) or masculine (e.g., plumbing, school policing, emergency response)—affects how instances of sexual harassment are perceived.[note 1] The aim of the current study is to determine if there are significant differences in the perception of sexual harassment incidents based on whether the job is categorized as masculine or feminine. We predicted that stereotypically gendered job types would impact the perceived validity of sexual harassment claims. Particularly, we predicted women in masculine jobs would have their claims interpreted less as sexual harassment compared to women in feminine jobs. We also predict that women in masculine jobs will be blamed more and be viewed as less prototypical women than women in feminine jobs.

## 2. Materials and Methods

### 2.1. Participants

A total of 427 participants were recruited online using CloudResearch’s CONNECT platform ([32]). To qualify for the study, participants had to be at least 18 years of age or older. Demographic information about the participants can be found in Table 1. All study materials and datasets can be found at https://osf.io/v7g9z/?view_only=2eb158908b70401685145e3b058e2d3e (accessed on 30 January 2025).

### 2.2. Procedures and Manipulation

The current study had a 2 (Sexual Harassment vs. No Sexual Harassment) × 2 (Feminine Job vs. Masculine Job) × 3 (Jennifer vs. Sara vs. Brenda) mixed methods design. After consenting to the study, participants were randomly assigned to read 3 different workplace situations. After reading a specific situation, participants completed the sexual harassment, blame victim, and prototypical woman measures for that specific vignette. Participants completed this process three times (one time for each victim: Jennifer, Sara, and Brenda). The job type and the presence of sexual harassment were randomly assigned for each victim vignette. After completing the study measures, participants were asked some follow-up questions and demographic information. Once they had finished the study, they were thanked for their participation and compensated USD2.00 to their CloudResearch account. The study took an average of 15 min to complete.

#### 2.2.1. Jennifer’s Manipulation

For the vignette involving Jennifer, participants read that she worked at a hotel and was either a cleaning staff member (feminine job condition) or a plumber (masculine job condition). They were told that Jennifer’s supervisor would either continuously ask her on dates (sexual harassment condition) or would ask her to complete work she did not feel comfortable completing (no sexual harassment condition).

#### 2.2.2. Sara’s Manipulation

For the vignette involving Sara, participants read that she worked at a school and was either a teacher (feminine job condition) or a school police officer (masculine job condition). They were told that Sara’s supervisor either groped her (sexual harassment condition) or accidentally tripped her (no sexual harassment condition).

#### 2.2.3. Brenda’s Manipulation

For the vignette involving Brenda, participants read that she worked at a hospital and was either a nurse (feminine job condition) or an emergency medical technician [EMT] (masculine job condition). They were told that Brenda’s manager either showed her a photo of a penis (sexual harassment condition) or showed her a photo of their new workplace uniform (no sexual harassment condition).

### 2.3. Measures

#### 2.3.1. Sexual Harassment Measure

The sexual harassment measure was adapted from [29] ([29]). The scale consisted of four items (e.g., “In my opinion, Jennifer/Sara/Brenda experienced sexual harassment from the supervisor”) on a 1 (Strongly Disagree) to 7 (Strongly Agree) scale (αJennifer = 0.89, αSara = 0.97, αBrenda = 0.97).

#### 2.3.2. Blame Victim Measure

The blame victim measure was adapted from [60] ([60]). The scale consisted of six items (e.g., “Jennifer/Sara/Brenda is responsible for what happened”) on a 1 (Strongly Disagree) to 7 (Strongly Agree) scale (αJennifer = 0.87, αSara = 0.85, αBrenda = 0.86).

#### 2.3.3. Prototypical Woman Measure

The prototypical woman measure was adapted from [29] ([29]). The scale consisted of five items (e.g., “Jennifer/Sara/Brenda has a lot in common with other women”) on a 1 (Strongly Disagree) to 7 (Strongly Agree) scale (αJennifer = 0.86, αSara = 0.87, αBrenda = 0.85).

### 2.4. Data Cleaning

For the current study, we cleaned the data in accordance with [57] ([57]). Individual missing items on scale measures were left as missing, and composite scores were created excluding those items. Participants missing complete scale measures were excluded using pairwise deletion. Outliers were assessed and were winsorized.

To test our specific predictions, 2 (Sexual Harassment vs. No Sexual Harassment) × 2 (Feminine Job vs. Masculine Job) Between Groups ANOVAs were conducted for each sexual harassment vignette participants read, based on the dependent variable of interest. Given that each manipulation only had two conditions, no pairwise post hoc tests for significant main effects were conducted.[note 2]

## 3. Results

### 3.1. Manipulation Checks

To assess that participants correctly read the vignettes, we asked participants what happened in the vignette for Jennifer, Sara, and Brenda. For Jennifer, the majority of participants in the sexual harassment condition (98.58%) and the no sexual harassment condition (96.70%) correctly recalled what they had read about. For Sara, the majority of participants in the sexual harassment condition (99.06%) and the no sexual harassment condition (99.52%) correctly recalled what they had read about. For Brenda, all of the participants in the sexual harassment condition and the majority in the no sexual harassment condition (97.12%) correctly recalled what they had read about.

To assess if participants viewed the specific jobs as feminine and masculine, participants were asked “How would you describe Jennifer’s/Sara’s/Brenda’s job field on the following traits” and were provided with a 7-point scale ranging from 1 (Feminine) to 7 (Masculine). Results can be found in Table 2. Overall, participants viewed the feminine conditions as more feminine than the masculine conditions (*p*’s < 0.001).

### 3.2. Jennifer Conditions Results

To assess the effect of job type and sexual harassment for the Jennifer conditions, a 2 (Sexual Harassment vs. No Sexual Harassment) × 2 (Feminine Job vs. Masculine Job) Between Groups ANOVA was conducted for the three variables separately. ANOVA results for the Jennifer conditions can be found in Table 3, and conditional means can be found in Table 4. For ease of interpretation, only significant results are described here. For all three measures, there was a significant main effect for the sexual harassment manipulation (*p*’s < 0.008). For the prototypical woman measure, there was also a significant main effect of job type (*p* < 0.001).

To assess if viewing Jennifer as a prototypical woman mediated the effects of the manipulations on the sexual harassment measure, a moderated mediation model was conducted in PROCESS using Model 7 ([33]). The job type manipulation was entered as the X variable, the sexual harassment manipulation was entered as the W variable, the prototypical woman measure was entered as the M variable, and the sexual harassment measure was entered as the Y variable (see Figure 1).

Although the prototypical woman measure was a significant mediator for the relationship between the job type manipulation and the sexual harassment measure for the sexual harassment condition [−0.272, −0.042] and not the no sexual harassment condition [−0.223, 0.012], the difference in this effect was not significant [−0.076, 0.196]. For people in the sexual harassment conditions, those viewing the feminine manipulation also viewed Jennifer as a more prototypical woman than those viewing the masculine condition (*p* = 0.001), and those viewing Jennifer as a more prototypical woman also viewed the situation as sexual harassment more than those viewing Jennifer as a less prototypical woman (*p* = 0.03). The mediation effect of the prototypical woman measure on the relationship between job type and the sexual harassment measure was not significant for the no sexual harassment conditions (see Figure 2).

### 3.3. Sara Conditions Results

To assess the effect of job type and sexual harassment for the Sara conditions, a 2 (Sexual Harassment vs. No Sexual Harassment) × 2 (Feminine Job vs. Masculine Job) Between Groups ANOVA was conducted for the three variables separately. ANOVA results for the Sara conditions can be found in Table 5 and conditional means can be found in Table 4. For ease of interpretation, only significant results are described here. For the sexual harassment and prototypical woman measures, there was a significant main effect for the sexual harassment manipulation (*p*’s < 0.008). For the prototypical woman measure, there was also a significant main effect of job type (*p* < 0.001).

To assess if viewing Sara as a prototypical woman mediated the effects of the manipulations on the sexual harassment measure, a moderated mediation model was conducted in PROCESS using Model 7 ([33]). The job type manipulation was entered as the X variable, the sexual harassment manipulation was entered as the W variable, the prototypical woman measure was entered as the M variable, and the sexual harassment measure was entered as the Y variable (see Figure 1).

The prototypical woman measure was a significant mediator for the relationship between the job type manipulation and the sexual harassment measure for the sexual harassment condition [−0.353, −0.047] and the no sexual harassment condition [−0.449, −0.066], but the difference in this effect was not significant [−0.241, 0.101]. For both people in the sexual harassment conditions and no sexual harassment, participants who viewed the feminine manipulation viewed Jennifer as a more prototypical woman than participants who viewed the masculine condition (*p*’s < 0.001), but participants’ view of Jennifer as a woman did not impact perceptions of sexual harassment (*p*’s > 0.08) (see Figure 3).

### 3.4. Brenda Conditions Results

To assess the effect of job type and sexual harassment for the Brenda conditions, a 2 (Sexual Harassment vs. No Sexual Harassment) × 2 (Feminine Job vs. Masculine Job) Between Groups ANOVA was conducted for the three variables separately. ANOVA results for the Brenda conditions can be found in Table 6 and conditional means can be found in Table 4. For ease of interpretation, only significant results are described here. For the sexual harassment measure, there was a significant main effect for the sexual harassment manipulation (*p* < 0.001).

Although we planned to assess if viewing Brenda as a prototypical woman mediated the effects of the manipulations on the sexual harassment measure, because there were no significant effects for the prototypical woman measure, we did not conduct the analyses.

## 4. Discussion

The current research examined how job or career types may impact perceptions of sexual harassment differently based on how prototypical people view a woman victim. For sexual harassment that involves unwanted romantic attention (i.e., the Jennifer conditions), working in masculine fields can impact how well a victim fits into the gender prototype a person has about women. The impact on perceptions of the woman can then lead to people having a harder time labeling her experiences as sexual harassment. A similar effect can happen when women experience sexual harassment in the form of unwanted sexual advances (i.e., the Sara conditions). Again, working in masculine fields can impact how well people believe a victim fits into the gender prototype of a woman and these perceptions of the woman can make it more difficult for people to label her experiences as sexual harassment. The current study suggests these barriers may not be present or as strong for all types of sexual harassment (e.g., gender harassment [the Brenda conditions]).

The impact of career type (masculine vs. feminine) on perceptions of a victim in the current study is consistent with past research on gender norms in the workplace ([19]; [41]). People often use societal norms regarding workplace behaviors to label a specific context as inappropriate or not ([19]). This tendency may be why sexual harassment in male-dominated workplaces is viewed as more normalized than in female-dominated workplaces ([34]; [7]). Past research on sexual harassment prototypes has also demonstrated that perceptions of sexual harassment and perceptions of prototypicality in a victim are heavily related ([29]).

The current research also suggests that the impact of prototypicality on perceptions of sexual harassment is not consistent across all types of sexual harassment. Specifically, being a prototypical woman does not seem to impact how people perceive gender harassment (e.g., being shown sexually explicit images). One reason for the lack of effect between prototypes and perceptions of sexual harassment for specific types of behaviors may relate more to workplace norms than gender norms. The workplace tends to have norms around humor and sexual conversations, usually prohibiting such behaviors for all employees ([38]; [53]). Workplace norms around dating behaviors and fraternization are not always as strict, many people use workplaces to find friends and romantic partners ([54]). Given the average person’s tendency to believe sexual harassment is linked to romantic attraction ([42]) and romantic behaviors in the workplace is somewhat normalized ([59]), it is realistic that people may use gender norm as a way to understand unwanted sexual or romantic attention based sexual harassment more than gender harassment based sexual harassment. Overall, the current study suggests that victims of sexual harassment may face unique challenges when reporting their experiences based on the type of sexual harassment and their current career path.

### 4.1. Implications for the Study

The results of this study imply a need for comprehensive education about sexual harassment. The minimization and dismissal of certain sexual harassment behaviors perpetuate and encourage incidences of sexual harassment in the workplace ([31]). Reducing tolerance of sexual harassment in the workplace can decrease the incidence of sexual harassment and increase reporting ([43]; [31]). Unfortunately, training interventions in the workplace have triggered a backlash and have been found ineffective for a long time ([44]). Educating students in universities and high schools about different types of sexual harassment and the negative effects of sexual harassment may be an alternative that could be successful in reducing sexual harassment incidence. Furthermore, having trainings that provide several examples of sexual harassment and victims can help broaden the prototype people have around sexual harassment.

Research has indicated that workplaces with near-equal gender ratios typically exhibit lower rates of harassment and tolerance of harassment compared to single-gender-dominated workplaces ([48]). If occupations that are primarily dominated by one gender were to experience a shift toward a greater gender balance, it is plausible that the incidence of harassment would decrease as a result ([36]; [48]). Additionally, expanding gender ratios in the workplace could also shift perceptions of gender norms in said workplace, which could help reduce the effect of gender norm violations on perceptions of sexual harassment. Addressing occupational inequalities is thus crucial to developing interventions that ensure all harassment claims are judged equitably, regardless of the victim’s gender, position, or job type.

Moreover, power dynamics in workplace hierarchies further complicate perceptions, as harassment from male supervisors towards women is often more prevalent, especially when women are in gender-minority positions ([21]). Organizational efforts to move away from masculinity contest cultures should benefit everyone and are especially likely to improve outcomes for women who defy traditional gender norms and are most affected by harassment, and discrimination ([23]; [36]). Equity and diversity in the workplace correlate with less harassment. Implementing policies such as DEI initiatives and cultivating a sense of belonging for everyone in the workplace could help mitigate harassment behaviors ([46]; [14]).

### 4.2. Limitations and Future Directions

There are several important limitations to note for the current research. First, participants were only shown short vignettes for each of the sexual harassment manipulations. Although the methods in the current study are similar to past studies ([29]), not having detailed information about the workplace, victim, or perpetrator may impact how people view sexual harassment behaviors. For example, past research that used more detailed sexual harassment manipulations (i.e., included interviews with victims/perpetrators and formal complaint information) did not find a relationship between prototype deviation and how prototypical participants viewed the victim ([40]) or participants’ perceptions of the sexual harassment claims. Future research should assess the external validity of the current findings by examining how the setting or job type impacts perceptions of sexual harassment in real-world cases or when people are given more information about the sexual harassment event.

A second important limitation around sexual harassment prototypes in the current study is the lack of photos for the participants to read about. By not showing an image of the victim herself, we do not know if the manipulation of job type had an impact on the mental image people had of the victim. For example, in the Jennifer conditions, we do not know if the decrease in prototypicality and perceptions of sexual harassment in the plumber condition was due to the masculine stereotypes of plumbers or because the mental image participants had of Jennifer in the plumber condition was physically more masculine (e.g., wider jaw, larger face) compared to the cleaning staff condition. Furthermore, some recent studies on sexual harassment have suggested that keeping images of victims consistent when manipulating sexual harassment behaviors may decrease the relationship between prototypicality and perceptions of sexual harassment ([40]). Future studies should examine how job type and manipulations around sexual harassment settings impact perceptions of sexual harassment, even when the victim’s physical characteristics are kept consistent.

Another limitation of this current study is that we did not assess participant’s personal experiences around sexual harassment or how those experiences impacted their perceptions. For example, victims of sexual harassment often report their own experiences impacting how they rationalize sexual harassment they witness from others ([11]). Given that people who experience sexual harassment often view other’s sexual harassment differently ([12]; [62]), not including a measure of sexual harassment experience may limit the external validity of the current study. Future research should attempt to assess how personal experiences around sexual harassment can impact the prototypes people have around sexual harassment victims, perpetrators, and situations.

## 5. Conclusions

Between 25 and 85% of people report experiencing sexual harassment ([20]; [45]), and sexual harassment costs companies USD1–USD2.2 billion in 2020 ([1]). The current study demonstrated that prototypes around gender and sexual harassment can result in differences in how people define sexual harassment. People who view women victims of sexual harassment as prototypical women tend to more easily label sexual harassment compared to people who view women victims as non-prototypical women. Although these effects may relate to the type of jobs a woman has, in general, gender norms in the workplace can directly relate to perceptions of sexual harassment. Future researchers and practitioners need to create interventions and trainings that help expand prototypes of women in the workplace so victims of sexual harassment do not have additional barriers based on where they work or the gender norms they may or may not violate.

## Figures and Tables

**Figure 1 behavsci-15-00757-f001:**
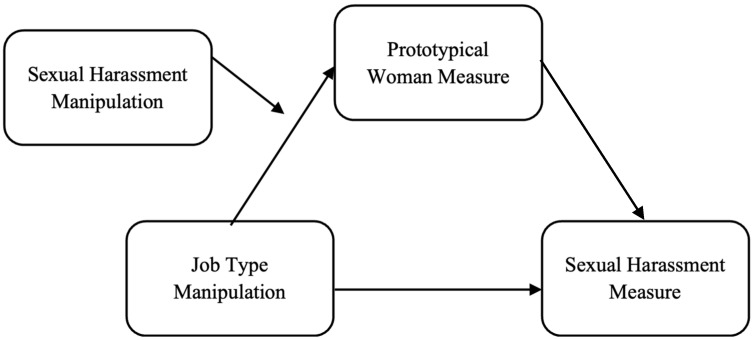
Moderated mediation model for all victims.

**Figure 2 behavsci-15-00757-f002:**
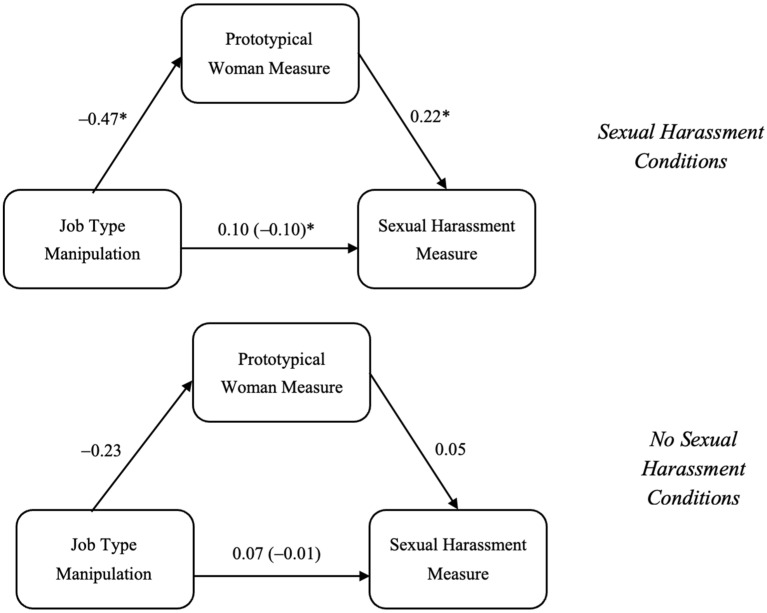
Mediation models based on sexual harassment manipulation for Jennifer. Note. * denotes significance.

**Figure 3 behavsci-15-00757-f003:**
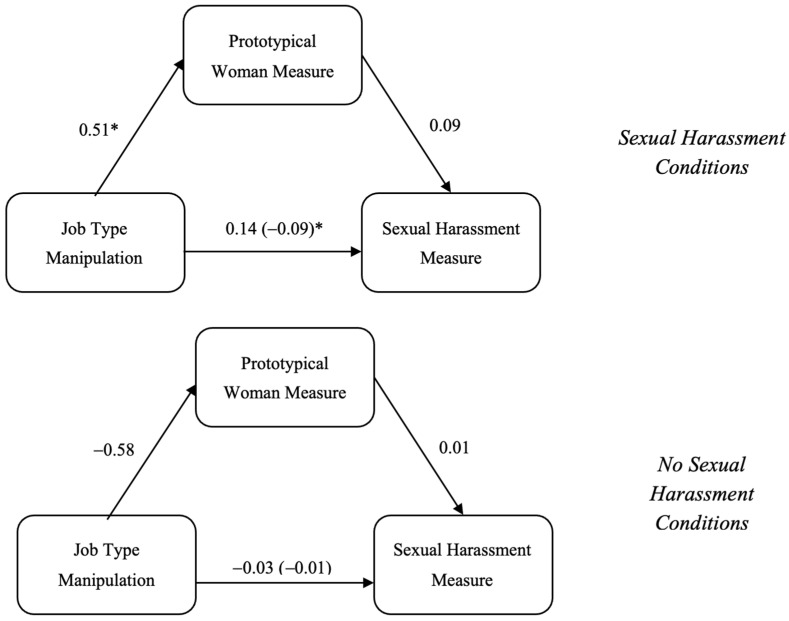
Mediation models based on sexual harassment manipulation for Sara. *Note.* * denotes significance.

**Table 1 behavsci-15-00757-t001:** Participants’ Demographic Information.

Demographic Variable	Number of Participants
Men	207
Women	199
Other Gender	8
White	272
Black/African American	55
Asian/Asian American	40
Indigenous Nation/Native American	1
Latinx/Hispanic American	30
Multiracial/Other Race	16
Age	37.03 (11.50)
Political Ideology	3.17 (1.75)

*Note*. Age (a continuous measure) was reported as *Mean* (*SD*). Political ideology (a continuous measure [1 = Strong Democrat; 7 = Strong Republican]) was reported as *Mean* (*SD*).

**Table 2 behavsci-15-00757-t002:** Manipulation Check Results.

	Feminine Condition *Mean* (*SD*)	Masculine Condition *Mean* (*SD*)	*t*-Test	Cohen’s *d* CI
Jennifer Conditions	3.55 (1.34)	4.91 (1.54)	9.75 *	[0.75, 1.15]
Sara Conditions	2.96 (1.28)	4.26 (1.41)	9.89 *	[0.76, 1.17]
Brenda Conditions	2.97 (1.29)	3.97 (1.27)	8.00 *	[0.58, 0.98]

*Note.* * *p* < 0.001.

**Table 3 behavsci-15-00757-t003:** ANOVA results for the Jennifer conditions.

	*F*-Test	*p*-Value	Partial Eta Squared
Sexual Harassment Measure			
Sexual Harassment Manipulation	603.84	<0.001	0.592
Job Type Manipulation	0.02	0.891	0.000
Interaction	0.02	0.890	0.000
Blame of Victim Measure			
Sexual Harassment Manipulation	11.64	<0.001	0.027
Job Type Manipulation	0.81	0.370	0.002
Interaction	0.01	0.929	0.000
Prototypical Woman Measure			
Sexual Harassment Manipulation	7.46	0.007	0.018
Job Type Manipulation	12.37	<0.001	0.029
Interaction	0.82	0.366	0.002

**Table 4 behavsci-15-00757-t004:** Main effect conditional means for all victims.

	Sexual Harassment Condition	No Sexual Harassment Condition	Feminine Condition	Masculine Condition
Jennifer Conditions				
Sexual Harassment Measure	5.32 (1.45)	2.23 (1.09)	3.76 (1.97)	3.78 (2.05)
Blame of Victim Measure	1.74 (1.00)	2.11 (1.21)	1.97 (1.09)	1.88 (1.16)
Prototypical Woman Measure	4.96 (1.03)	4.67 (1.17)	5.00 (1.00)	4.63 (1.19)
Sarah Conditions				
Sexual Harassment Measure	6.63 (0.78)	1.66 (1.03)	4.15 (2.63)	4.19 (2.67)
Blame of Victim Measure	1.50 (0.78)	1.43 (0.95)	1.42 (0.84)	1.51 (0.89)
Prototypical Woman Measure	5.15 (1.06)	4.88 (1.06)	5.28 (0.99)	4.74 (1.10)
Brenda Conditions				
Sexual Harassment Measure	6.50 (0.98)	1.61 (1.14)	4.10 (2.64)	4.08 (2.70)
Blame of Victim Measure	1.50 (0.84)	1.66 (1.07)	1.52 (0.88)	1.64 (1.04)
Prototypical Woman Measure	5.18 (0.93)	5.04 (0.93)	5.17 (0.97)	5.06 (1.02)

**Table 5 behavsci-15-00757-t005:** ANOVA results for the Sara conditions.

	*F*-Test	*p*-Value	Partial Eta Squared
Sexual Harassment Measure			
Sexual Harassment Manipulation	3097.64	<0.001	0.881
Job Type Manipulation	0.11	0.737	0.000
Interaction	0.93	0.335	0.001
Blame of Victim Measure			
Sexual Harassment Manipulation	0.54	0.462	0.001
Job Type Manipulation	1.15	0.285	0.003
Interaction	0.03	0.875	0.001
Prototypical Woman Measure			
Sexual Harassment Manipulation	7.42	0.007	0.018
Job Type Manipulation	28.81	<0.001	0.065
*Interaction*	0.41	0.521	0.001

**Table 6 behavsci-15-00757-t006:** ANOVA results for the Brenda conditions.

	*F*-Test	*p*-Value	Partial Eta Squared
Sexual Harassment Measure			
Sexual Harassment Manipulation	2199.67	<0.001	0.842
Job Type Manipulation	0.06	0.814	0.000
Interaction	0.16	0.694	0.000
Blame of Victim Measure			
Sexual Harassment Manipulation	2.87	0.091	0.007
Job Type Manipulation	1.42	0.234	0.003
Interaction	0.11	0.740	0.000
Prototypical Woman Measure			
Sexual Harassment Manipulation	1.96	0.162	0.005
Job Type Manipulation	1.28	0.258	0.003
Interaction	1.25	0.265	0.003

## Data Availability

The raw data supporting the current research can be found at https://osf.io/v7g9z/?view_only=2eb158908b70401685145e3b058e2d3e (accessed on 30 January 2025).

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
