# Peer review of "Do People Judge Sexual Harassment Differently Based on the Type of Job a Victim Has?"

_behavsci, 2025, doi:10.3390/bs15060757_

Round 1
Reviewer 1 Report
Comments and Suggestions for Authors
Based on a sample collected online, the current study relates sexual harassment judging to the specific job types of victims. The study demonstrated that prototypes around gender and sexual harassment can result in differences in how people define sexual harassment. People who view women victims of sexual harassment as prototypical women tend to more easily label sexual harassment compared to people who view women victims as non-prototypical women. In my personal view, the manuscript is carefully designed and already well-written, making it much better in quality than 95% of the manuscripts I reviewed for the current journal.
I propose a few minor suggestions, mainly for clarification purposes.
1. The study is based on a sample of 427 adults. Demographic information such as age, gender, race, and political ideology has been collected and demonstrated. Why not further collect several other critical demographic factors such as education and marital status, etc.? More importantly, this demographic information should be used for either of the two purposes: 1) showing that the demographic distribution in the sample is similar to that of the target population it aims to represent, thereby supporting credible statistical inferences. 2) They should be used as control variables in statistical models, such as the mediation models in the current study.
2. Between the section measures and the section results, there should be a subsection following the measures section, naming analytical steps or analytical strategy. This section will delineate the specific steps (first, second, ...) and methods (e.g., anova, what kind of test, what mediation approach exactly...) used to illustrate the results so that readers can more easily follow the progress of the results reporting.
Author Response
- The study is based on a sample of 427 adults. Demographic information such as age, gender, race, and political ideology has been collected and demonstrated. Why not further collect several other critical demographic factors such as education and marital status, etc.? More importantly, this demographic information should be used for either of the two purposes: 1) showing that the demographic distribution in the sample is similar to that of the target population it aims to represent, thereby supporting credible statistical inferences. 2) They should be used as control variables in statistical models, such as the mediation models in the current study.
Author Response: The comment about the demographics is very helpful and the reviewer brings up great points that other demographics may have been useful. Due to funding and time restraints, we only collected information on the demographics reported. Reviewer’s second comment about controlling for the demographics, we did not predict that any specific demographic would impact the results, therefore, we did not control for those factors. The goal of the demographics was to assess broad generalizability based on sample factors. That being said, we did run additional analyses based on the reviewer’s comments and the overall results did not change when we controlled for demographics.
- Between the section measures and the section results, there should be a subsection following the measures section, naming analytical steps or analytical strategy. This section will delineate the specific steps (first, second, ...) and methods (e.g., anova, what kind of test, what mediation approach exactly...) used to illustrate the results so that readers can more easily follow the progress of the results reporting.
Author Response: Thank you for this suggestion. We have added information about data cleaning and how ANOVAs were conducted. We agree that this information for make the results sections easier to follow.
Reviewer 2 Report
Comments and Suggestions for Authors
I would like to commend the authors for their work on sexual harassment in the workplace. This study addresses an important issue in the workplace and makes a valuable contribution to our understanding of sexual harassment in the workplace. Overall, the article is well-written and the research design is sound. There are a few areas where I believe improvements can be made to strengthen the paper further, which I have outlined below.
Literature Review
In the literature review, I would like to see more discussion of different forms of sexual harassment in the workplace. On lines 29-31, it is mentioned, but there does not appear to be specific research that is outlined that describes the differences in workplaces. Specifically, across industries. This would also help with the reasoning behind why the authors chose the specific jobs for the vignettes.
Lines 103-115, there is a little talk about intersectionality between gender and race and the prototype theory. More discussion would be nice here on what the theory states regarding race and gender and harassment.
Current Study
Why were the specific jobs picked for the vignettes? Was that personal preference or is there research that indicates these are the most masculine jobs/jobs with the most harrasment?
Under the measures section, more information about the scales would be nice. This is a very interesting part of the research so having more information about the scales seems warranted (even as an appendix).
Author Response
Response to Reviewer 2 Comments
|
|
1. Summary |
|
Thank you very much for taking the time to review this manuscript. Please find the detailed responses below and the corresponding revisions/corrections highlighted/in track changes in the re-submitted files
|
|
2. Questions for General Evaluation |
Reviewer’s Evaluation |
Does the introduction provide sufficient background and include all relevant references? |
Yes |
Are all the cited references relevant to the research? |
Yes |
Is the research design appropriate? |
Yes |
Are the methods adequately described? |
Yes |
Are the results clearly presented? |
Yes |
Are the conclusions supported by the results? |
Yes |
3. Point-by-point response to Comments and Suggestions for Authors |
|
Comments 1: In the literature review, I would like to see more discussion of different forms of sexual harassment in the workplace. On lines 29-31, it is mentioned, but there does not appear to be specific research that is outlined that describes the differences in workplaces. Specifically, across industries. This would also help with the reasoning behind why the authors chose the specific jobs for the vignettes. |
|
Response 1: We agree that more information about workplace specifc harassment makes our opening argument stronger. Thank you for pointing this issue out. We have added information and references on line 31. “For example, Hibino and colleagues (2008) reported that 56% of women nurses had experienced sexual harassment in the workplace, but Schlick and colleagues (2021) reported 80% of women medical residents experienced gender based sexual harassment, and 43% experienced unwanted sexual or romantic attention. Specific career paths can also create barriers around reporting and addressing sexual harassment; women police officers report threats of retaliation and dismissive attitudes towards sexual harassment in the workplace (Davis et al., 2023). Furthermore, Riddle and Heaton (2023) theorize that women working in male-dominated fields may be more likely to experience sexual harassment due to the increased sexualization and feminization of women in those fields.” |
|
Comments 2: Lines 103-115, there is a little talk about intersectionality between gender and race and the prototype theory. More discussion would be nice here on what the theory states regarding race and gender and harassment. |
|
Response 2: Author note: Intersectionality is an important factor for this research, so we agree more information is needed. We adding the following context to tie Prototype theory back to race and gender: “Prototype theory suggests that the tendency to dismiss Black victims of sexual harassment more than White victims may stem from a mismatch between the societal prototype of a sexual harassment victim being a “typical woman “(i.e., a White woman; Goh et al., 2020) and the race of the specific victim (i.e., a Black woman).” |
|
Comments 3: Why were the specific jobs picked for the vignettes? Was that personal preference or is there research that indicates these are the most masculine jobs/jobs with the most harassment? |
|
Response 3: We added a footnote to explain more about the jobs/careers in the materials. Basically, we piloted a series of different jobs (e.g., teacher, actor, accountant) and then matched careers that were considered highly masculine with careers that were highly feminine. For example, police officer was considered a masculine job and teacher was considered a feminine job. Therefore, we manipulated teacher and school officer to the setting itself (Simon and Smith Elementary) could remain consistent between conditions. |
|
Comments 4: Under the measures section, more information about the scales would be nice. This is a very interesting part of the research so having more information about the scales seems warranted (even as an appendix). |
|
Response 4: We agree that details about the measures are important for the paper. We have a hyperlink in the methods section now that will take readers to the study materials so that they can see the specific measures (and overall study materials). Thank you for the suggestion. |